

# Identification of proteins related to SIS3 by iTRAQ and PRM-based comparative proteomic analysis in cisplatin-induced acute kidney injury

Jiayan Huang[1,2,*], Jian Ye[2,*], Yi Gao[3], Yu Wang[2], Qing Zhao[2], Tanqi Lou[1] and Weiyan Lai[1]

[1] Department of Nephrology, The Third Affiliated Hospital of Sun Yat-sen University, Guangzhou, Guangdong, China
[2] Department of Nephrology, The First Affiliated Hospital, Jiangxi Medical College, Nanchang University, Nanchang, Jiangxi, China
[3] Department of Critical Care Medicine/ICU (Intensive Care Unit), The Second Affiliated Hospital, Jiangxi Medical College, Nanchang University, Nanchang, Jiangxi, China
[*] These authors contributed equally to this work.

Corresponding author
Weiyan Lai, laiwy5@mail.sysu.edu.cn

## ABSTRACT

**Background.** Cisplatin is a commonly used nephrotoxic drug and can cause acute kidney injury (AKI). In the present study, isobaric tags for relative and absolute quantification (iTRAQ) and parallel reaction monitoring (PRM)-based comparative proteomics were used to analyze differentially expressed proteins (DEPs) to determine the key molecular mechanism in mice with cisplatin-induced AKI in the presence or absence of SIS3, a specific p-smad3 inhibitor, intervention.

**Methods.** The cisplatin-induced AKI mouse model was established and treated with SIS3. We used iTRAQ to search for DEPs, PRM to verify key DEPs and combined Gene Ontology (GO) and Kyoto Encyclopedia of Genes and Genomes (KEGG) for bioinformatics analysis. We then assessed lipid deposition, malondialdehyde (MDA) and reactive oxygen species (ROS) and detected the expression of SREBF1, SCD1, CPT1A, PPARα and NDRG1 in vitro.

**Results.** Proteomic analysis showed that the identified DEPs were mainly enriched in energy metabolism pathways, especially in lipid metabolism. When SIS3 was applied to inhibit the phosphorylation of Smad3, the expression of NDRG1 and fatty acid oxidation key proteins CPT1A and PPARα increased, the expression of lipid synthesis related proteins SREBF1 and SCD1 decreased and the production of lipid droplets, MDA and ROS decreased.

**Conclusion.** SIS3 alleviates oxidative stress, reduces lipid accumulation and promotes fatty acid oxidation through NDRG1 in cisplatin-induced AKI. Our study provides a new candidate protein for elucidating the molecular mechanisms of fatty acid metabolism disorders in cisplatin-induced acute kidney injury.

# INTRODUCTION

Acute kidney injury (AKI) is a clinical syndrome caused by acute renal dysfunction in a short period of time, and is an important factor leading to chronic kidney disease (CKD) and a public health issue affecting the survival of millions of patients worldwide (*Kellum et al., 2021*). AKI is caused by a variety of factors, about one-fifth of cases are caused by acute tubular necrosis due to nephrotoxic drugs, including the most common cancer drug-cisplatin (*Wen et al., 2013*). Cisplatin is a highly effective anticancer drug, but its clinical application is limited due to its severe renal toxicity (*McSweeney et al., 2021*). The pathophysiological mechanisms of cisplatin-induced AKI are inflammation, oxidative stress, vascular damage, DNA damage, mitochondrial dysfunction and apoptosis (*Yang et al., 2014*; *Zhu et al., 2015*). To date, there are still no effective drugs to prevent or treat cisplatin-induced nephrotoxicity. Therefore, it is crucial to understand the molecular mechanism to identify an effective intervention to prevent cisplatin-induced AKI.

In recent years, more and more researchers have used proteomics to identify and quantify proteins in different cells or tissues and explore disease-related processes. Through proteomics combined with bioinformatics analysis, molecules that play a key role in the pathogenesis of diseases can be predicted. Recently, many potential diagnostic and therapeutic markers have been identified in AKI. Human β-defensin-1 has been found to exert renal protective effects in contrast-induced AKI (*Bennett et al., 2008*). Researchers have discerned the increased levels of albumin, α-1 antitrypsin, and β-2 microglobulin, along with decreased levels of fibrinogen α-chain and collagen fragments types 1α(I) and 1α(III), in cases of AKI. This was achieved through capillary electrophoresis-mass spectrometry analysis of urine specimens from a cohort of 87 ICU patients. The detected patterns of upregulation and downregulation present a significant diagnostic potential for the early identification of AKI (*Metzger et al., 2010*). Using contemporary shotgun proteomics to analyze urine samples from mice with sepsis, B. Maddens and colleagues highlighted chitinase-like proteins as promising markers for diagnosing sepsis-induced AKI. Specifically, they identified urinary chitinase 3-like protein 1, chitinase 3-like protein 3, and acidic mammalian chitinase as effective indicators of sepsis-induced AKI in mice. Nonetheless, additional research is needed to enhance our comprehension of the role these proteins play in human subjects (*Maddens et al., 2012*). Furthermore, recent research has indicated that patients with acute-on-chronic liver failure who exhibit a plasma metallothionein concentration greater than 5.83 ng/mL face a heightened risk of developing AKI (*Acharya et al., 2022*).

The TGF-β/Smad signaling pathway plays a key role in various physiological and pathological processes, including cell growth and differentiation, apoptosis, inflammatory response, cell cycle, and so on (*Massagué, 2012*). Smad3 is an important signal transduction and transcription regulatory factor in the TGF-β signaling pathway, which plays an important role in liver, kidney and heart fibrosis (*Hu et al., 2018*; *Wu et al., 2022*). Disruption of the TGF-β signaling pathway has been linked to a wide range of diseases such as cancer, bone disease, CKD and AKI (*Hu et al., 2018*). Recent research has found

that SARS-CoV-2 N protein is a key mediator for AKI and induces AKI *via* the Smad3-dependent G1 cell cycle arrest mechanism (*Wang et al., 2022*). Smad3 promotes AKI susceptibility in diabetic mice by interacting with p53 and NOX4 (*Wang et al., 2020*). We have demonstrated that Smad3 may have roles in cisplatin nephropathy due to its potential effects on tubular epithelial cell death and regeneration, and SIS3, a specific inhibitor of Smad3 protects against renal dysfunction and tubular necrosis and eliminates the exacerbation of cisplatin-induced AKI (*Huang et al., 2022*). However, the mechanism of Smad3 mediated cisplatin-induced AKI still requires further investigation. The use of proteomic techniques to explore the differential proteins associated with Smad3 can help us identify the different molecular mechanisms that regulate the occurrence of AKI.

The kidney filters a huge amount of blood every day and is a high energy consuming organ. Fatty acid oxidation (FAO) is the main energy metabolism mode of renal tubular epithelial cells (TECs) (*Houten et al., 2016*). Under various stimuli such as oxidative stress, inflammation, ischemia and hypoxia, and nephrotoxic substances, the mitochondrial respiratory chain function in renal TECs is significantly inhibited, resulting in impaired FAO function, increased deposition of free fatty acids, lipid peroxidation under attack by oxygen free radicals, and increased oxidative stress leading to lipotoxicity, ultimately leading to the occurrence of AKI (*Yin, Xu & Porter, 2011*; *Houten et al., 2016*). During AKI, some proteins are involved in energy production, including numerous peroxisomal matrix proteins that function in FAO (*Burton et al., 2023*). Studies have shown that there are significant FAO disorders and lipid deposition in the mouse AKI model induced by cisplatin (*Kamijo et al., 2007*). TGF-β/Smad3 is involved in the regulation of FAO in renal TECs. TGF-β may directly regulate the transcription level of PPARGC1a through Smad3, thus inhibiting FAO (*Kang et al., 2015*). Smad3 plays an important role in lipid metabolism-related kidney diseases, but whether Smad3 participates in fatty acid metabolism in cisplatin-induced AKI has not been reported.

In this study, as reported in our previous preprint (*Huang et al., 2023*), isobaric tags for relative and absolute quantification (iTRAQ) combined with high performance liquid chromatography-mass spectrometry (HPLC-MS/MS) was used to determine the distribution of protein abundance in the kidney of mice with cisplatin-induced AKI in the presence or absence of SIS3 intervention, and differentially expressed proteins (DEPs) were identified. Bioinformatics analysis was applied to explore the enriched functions and signaling pathways of DEPs, and parallel reaction monitoring (PRM) was applied to validate several overlapping DEPs. We then verified Smad3 regulated lipid metabolism and related proteins *in vitro*, and provided a new candidate protein for studying the fatty acid oxidation defects in cisplatin-induced AKI.

# MATERIALS & METHODS

## Animals

Twenty-four specific pathogen-free (SPF)-grade male SV129 mice (10–12 weeks old, 20–25 g) were obtained from the Vital River Laboratory Animal Technology Co. (Beijing, China) and maintained in a controlled environment (temperature: 20–25 °C, humidity: 45–55%)

in polypropylene cages (three mice per cage) with 12 h light-dark cycle. The experimental protocol was the Institutional Animal Care and Use Committee of the First Affiliated Hospital of Nanchang University (approval number: CDYFY-IACUC-202308QR037, Nanchang, China). Mice were allowed free access to standard rodent diet and drinking water. Mice were randomly divided into four groups after one-week: control group (Control), Smad3 inhibitor group (SIS3), cisplatin group (Cisplatin), and cisplatin+Smad3 inhibitor group (Cisplatin+SIS3). The control group was intraperitoneally injected with DMSO for 7 consecutive days. The cisplatin group was intraperitoneally injected with DMSO for 7 consecutive days, and intraperitoneally injected with cisplatin ($20\mu g/g$) on the 5th day. The SIS3 group was intraperitoneally injected with SIS3 ($5\mu g/g/day$) for 7 consecutive days. The cisplatin+SIS3 groups were intraperitoneally injected with SIS3 ($5\mu g/g/day$) for 7 consecutive days, and cisplatin ($20\mu g/g$) was intraperitoneally injected at 1 h after the injection of SIS3 on the 5th day. The health of each mouse was observed during the experiment. All mice were given drugs with an intraperitoneal injection and euthanized through isoflurane anesthesia. After humanely euthanized, the kidney and blood samples of the mice were harvested for subsequent analysis. Before the end of experiment, any animal showing persistent self-harm behavior or signs of unexpected disease will be euthanized immediately.

## iTRAQ LC-MS/MS based proteomic analysis

The proteins of mouse kidney tissue were quantified using the Bradford method and digested into peptides by filtration assisted proteome preparation (FASP). The peptide mixture of each sample was labeled with iTRAQ reagent. The peptide mixture was separated by high performance liquid chromatography (HPLC) and then analyzed by tandem mass spectrometry (LC-MS/MS). The peptide segments were detected by a mass spectrometer with the following specific parameters. Primary mass spectrometry parameters: resolution: 70,000, AGC target: 3e6, maximum IT: 100 ms, scan range: 350 to 1,800 m/z. Secondary mass spectrometry parameters: resolution: 17,500, AGC target: 5e4, maximum IT: 120 ms, TopN: 20, NCE/stepped NCE: 30. The RAW format raw mass spectrometry data obtained through a mass spectrometer will be imported in its entirety into ProteomeDiscoverer 1.4 software (version 1.4.0.288; Thermo Fisher, Waltham, MA, USA) and converted to the desired MGF format. Subsequently, the ProteinPilot™ Software 4.5 (version 1656, ABSciex) will be employed for online mass spectrometry database searching. The database for searching is https://www.uniprot.org/taxonomy/10090. The screening criteria for trusted proteins was false discovery rate (FDR) less than 0.01. The screening criteria for DEPs were changes greater than 1.2 times or less than 0.83 times, with a $p$-value less than 0.05.

## Validate DEPs by PRM

For PRM analysis, the protein was treated in the same way as iTRAQ. The proteins for verification were searched in the database, and then the peptides of these proteins were analyzed by skyline software to screen the target proteins suitable for PRM analysis. Finally, the peptide information was imported into Xcalibur software to set PRM test parameters. The peptides were separated by HPLC system, and then directly entered the mass spectrometer for on-line detection.

## GO and KEGG analysis

For the results of iTRAQ, basic statistical analysis, such as peptide coverage distribution and peptide number distribution, was performed first. Then the QuickGO online website (http://www.ebi.ac.uk/QuickGO/) was used to annotate the protein function for GO analysis. KEGG (http://www.genome) was used to analyze the enrichment pathway of DEPs. Hypergeometric test was used to calculate the results of GO annotation and KEGG pathway, and $p$ value less than 0.05 was significant.

## Protein–protein interaction networks (PPI) analysis

The interaction network between DEPs was mapped through STRING database (https://string-db.org/), screening protein interaction pairs with a comprehensive score greater than 0.7 for online analysis. The Cytoscape software (http://www.cytoscape.org/, version 3.9.1) was used to visualize protein synergistic effects, using the MCODE (Molecular Complex Detection) plug-in in Cytascape to filter key modules (Node Score Cutoff = 0.2, K-Core = 2, Max. Depth = 100).

## Cell culture

The mouse renal tubular epithelial cells(mTECs) were donated by Professor Lan Huiyao from the Li Ka Shing Institute of Health Sciences, The Chinese University of Hong Kong. The mTECs (ATCC, Manassas, VA, USA) were commercially available, which were passaged twice, then maintained in the lab. The mTECs were cultured in DMEM-F12 medium containing 10% fetal bovine serum (Clark, USA), 100 U/mL penicillin and 100 mg/mL streptomycin in a 5% $CO_2$/95% air atmosphere at 37 °C. After the cells grew to 70%–80% density, they were stimulated with cisplatin (5 μmol/L) in the presence or absence of SIS3 (5 μmol/L). After 24 h, the cells were collected for further experiments.

## Cell oil red O staining

Cells were fixed with 4% paraformaldehyde, dehydrated with isopropanol, and incubated in Oil Red O solution (Beyotime Biotechnology, Shanghai, China) in the dark for 30 min. After several washing steps, 60% isopropyl alcohol was added to redissolve the oil red O solution. Then, the nucleus was stained with hematoxylin for 3 min, photographs were taken with an optical microscope, and the collected images were quantitatively analyzed by Image Pro Plus6.0 software.

## Bodipy493/503 staining

The distribution of lipid droplets in cells was also observed by Bodipy 493/503 (MCE, Monmouth Junction, NJ, USA) staining. Cells were fixed at room temperature for 30 min with 4% paraformaldehyde, then washed with PBS, and incubated at 37 °C in darkness for 15 min with Bodipy493/503 at a final concentration of 2 umol/L. 4′,6-diamidino-2-phenylindole (DAPI) stained the nucleus for 5 min, and images were collected using a Zeiss inverted fluorescence microscope.

## Western blot

The proteins of the mTECs were extracted using the RIPA lysis buffer, and the protein concentration was then determined through the BCA method. Subsequently, the protein
samples were electrophoresed in a 10% SDS-PAGE gel and transferred onto PVDF membranes (Millipore, Burlington, MA, USA). Blocking of non-specific binding sites was performed using 5% skim milk for 1 h. Following this, the primary antibody was added for overnight incubation at 4 °C. The secondary antibody, labeled with horseradish peroxidase, was incubated for 1 h. Finally, the chemiluminescence imaging system was used for scanning, and the Image J software was utilized for semi-quantitative analysis. The primary antibodies used in this study include rabbit anti-Smad3 (ab40854; Abcam, Cambridge, UK), rabbit anti-P-Smad3 (ab52903; Abcam, Cambridge, UK), rabbit anti-SREBF1 (14088-1-AP; Proteintech, Rosemont, IL, USA), rabbit anti-SCD1 (ab236868; Abcam, Cambridge, UK), rabbit anti-CPT1A (15184-1-AP; Proteintech, Rosemont, IL, USA), rabbit anti-PPARα (15540-1-AP; Proteintech, Rosemont, IL, USA), rabbit anti-NDRG1 (ab124689; Abcam, Cambridge, UK), and rabbit anti-GAPDH (GB11002; Servicebio, Seattle, WA, USA).

### MDA and ROS assay

The MDA content in mTECs was detected by MDA Assay Kit (Beyotime Biotechnology, Shanghai, China). This method was based on the thiobarbituric acid method, and the specific operation was strictly in accordance with the kit instructions. The results were expressed in nmol MDA/mg protein. According to the instructions of the ROS Assay Kit (Beyotime Biotechnology, Shanghai, China), after incubation with DCF dye, the production of ROS in mTECs was detected by flow cytometry and observed by fluorescence microscopy.

### Immunohistochemical

For immunohistochemical staining of NDRG1, mouse kidney paraffin sections were antigens retrieved at high temperature and high pressure for 3 min, and then cooled and incubated with 3% hydrogen peroxide solution for 10 min to remove endogenous peroxidase. 10% goat serum was incubated at room temperature for 30 min, and primary antibody was added overnight at 4 °C. Secondary antibody was added, incubated at 37 °C for 30 min, and stained with DAB for 1min. Hematoxylin stained the nucleus. The images were calculated using the ImageJ software to calculate the average optical density value, that is, the concentration per unit area of positive staining.

### Statistical analysis

Graphpad Prism 8.0 software was used for statistical analysis. Measurement data are expressed as mean ± standard error. Independent sample $t$-test is used for comparison between the two groups. Two-way ANOVA with multiple comparisons is used for comparison between multiple groups. $P < 0.05$ indicates a statistically significant difference.

## RESULTS

### Protein profiling

Through iTRAQ quantitative proteomics analysis, we identified a total of 4787 proteins. Among them, 49.62% of the proteins had peptide segment coverage exceeding 10%

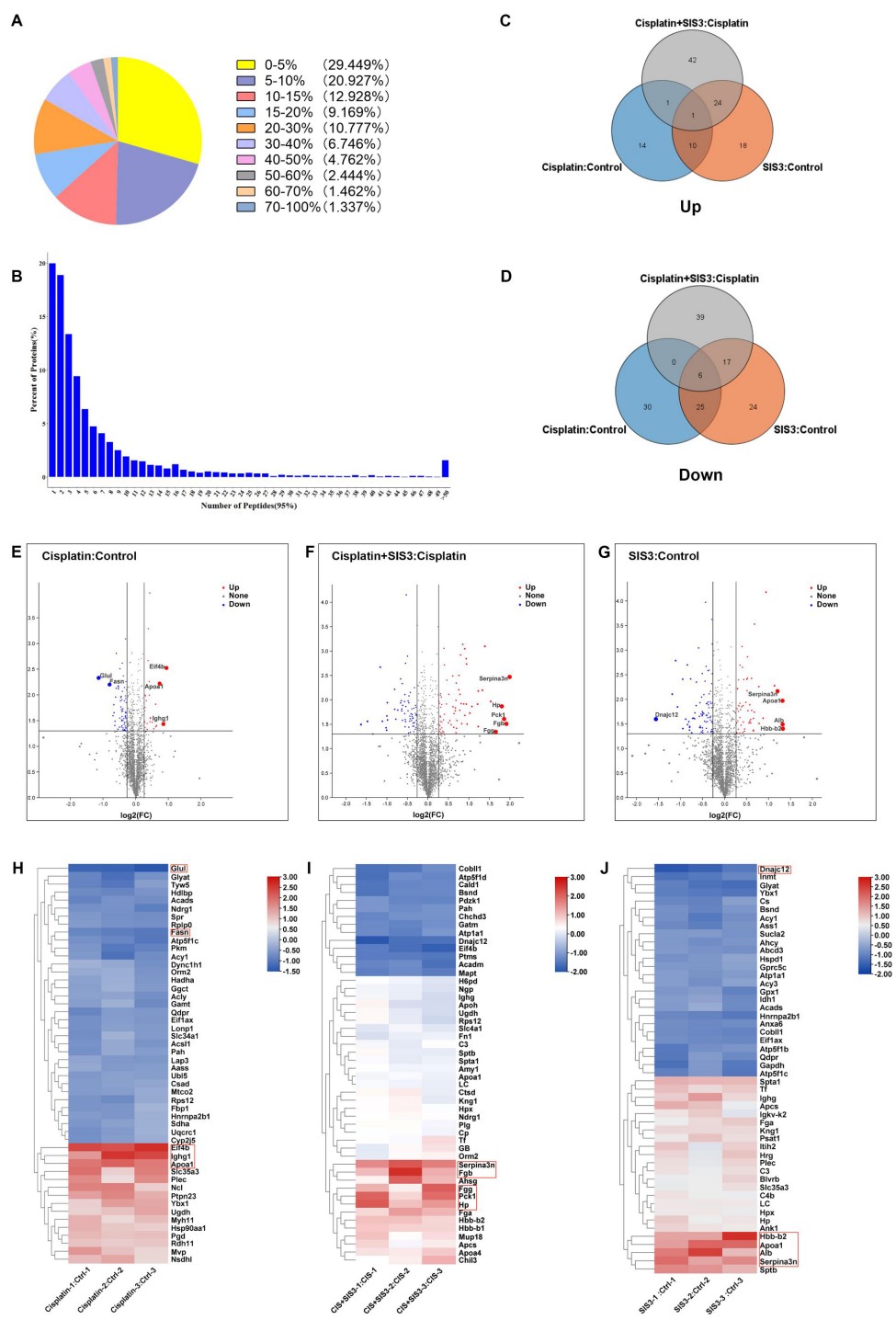

**Figure 1   Results of quantitative proteomic analysis screened from iTRAQ.** (A) Peptide sequence coverage distribution of proteins. (B) Quantity distribution of peptides. (C–D) Venn diagram of up-regulated and down-regulated DEPs. (E–G) Volcano plot for each group with the top five DEPs labeled. Statistically significant proteins are shown in the upper region, red dots are up-regulated proteins, blue dots are down-regulated proteins, and black dots indicate no significant differences. (H–J) Heat map of the hierarchical clustering results, with each row representing a protein and each column representing a set of samples. The top five DEPs are highlighted in the red box.

(Fig. 1A), and over 50% of the proteins contained three or more peptides (Fig. 1B), indicating that the identified proteins had good sequence coverage. As a result, 87 DEPs were found in the cisplatin group compared to the control group, of which 26 proteins were upregulated and 61 proteins were downregulated (Table S1). A total of 125 DEPs were found in the SIS3 group compared to the control group, of which 53 proteins were upregulated and 72 proteins were downregulated (Table S2). Compared with the cisplatin group, we found a total of 130 DEPs in the cisplatin+SIS3 group, of which 68 proteins were upregulated and 62 proteins were downregulated (Table S3). The Venn diagram showed the overlapped number of DEPs in each group (Figs. 1C and 1D), the figure showed two different protein trends among three comparison groups. The volcano plot showed the significant differences in proteins in the three groups (Figs. 1E, 1F and 1G). Simultaneously, the top 50 DEPs in each group were subjected to hierarchical clustering analysis (Figs. 1H, 1I and 1J).

## Functional and pathway analysis

We annotated the function of DEPs using the GO annotation tool QuickGO. Each protein was annotated from three aspects, cellular component, biological process, and molecular function. The hypergeometric test method was used to identify GO items that were significantly enriched in DEPs. Through GO analysis, we found that in biological processes, these proteins are mainly involved in cellular metabolic processes, biological regulation, and response to stimuli. With regard to cellular localization, these proteins are mainly distributed in cell structures, intracellular, and protein complexes. Among the molecular functions, the proteins involved in binding and catalytic activities were greatest (Figs. 2A–2C). Next, we analyzed the KEGG pathway of DEP$_S$ to determine the major biochemical, metabolic and signal transduction pathways in which they participate. Among the top 20 pathways enriched in DEPs, we found that most of these enrichment pathways were related to energy metabolism. The pathways enriched in the cisplatin group compared to the control group include fatty acid metabolism, carbon metabolism, biosynthesis of amino acids, the PPAR signaling pathway, oxidative phosphorylation and so on (Fig. 2D). The pathways enriched in the SIS3 group compared to the control group included carbon metabolism, the citrate cycle, the PPAR signaling pathway, fat digestion and absorption, cholesterol metabolism and so on (Fig. 2E). After SIS3 treatment, compared with the cisplatin group, the enriched pathways included the citrate cycle, carbon metabolism, amino acid biosynthesis, peroxisome, cholesterol metabolism and so on (Fig. 2F).

## PPI networks analysis

The STRING database was used to construct the interaction network between DEPs in each group. The Cytoscape plugin MCODE was used to detect the potential key modules and identify the top three scoring modules in each of the three groups. The module with the highest score had seven nodes and 20 edges in the cisplatin group compared to the control group (Fig. 3A). According to KEGG analysis, these proteins were mainly enriched in the fatty acid metabolism pathway. The module with the highest score had 10 nodes and 38 edges in the cisplatin+SIS3 group compared to the cisplatin group (Fig. 3B), and
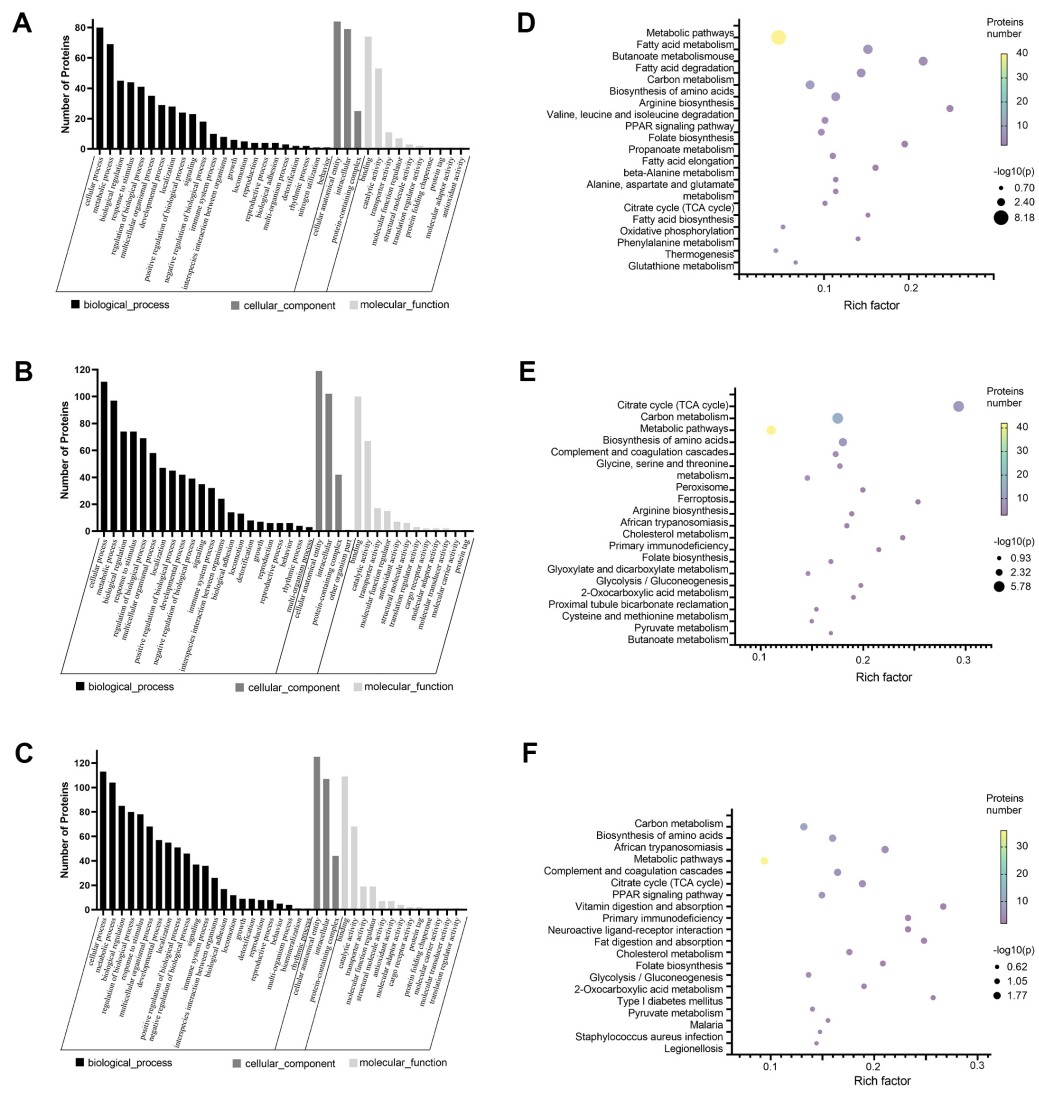

**Figure 2 GO and KEGG pathway enrichment analysis of the DEPs.** (A–C) GO annotation results. The abscissa represents GO annotation information, including biological processes, cell components, and molecular functions. The ordinate represents the number of proteins under each functional classification. (D–F) Bubble diagrams of the top 20 pathways. The abscissa is the enrichment factor, and the *p*-value is represented by dots of different colors. The change in color intensity represents different degrees of enrichment. The size of the dots represents the number of DEPs enriched in this pathway; the vertical axis is the name of each enrichment pathway. Cisplatin group *vs* control group (A, D). SIS3 group *vs* control group (B, E). Cisplatin+SIS3 group *vs* cisplatin group (C, F).

were significantly enriched in the digestion and absorption of vitamins and fats, and the cholesterol metabolism pathway. In the SIS3 group compared to the control group, the module with the highest score had nine nodes and 25 edges (Fig. 3C), and the DEPs were mainly enriched in the cholesterol metabolism pathway.

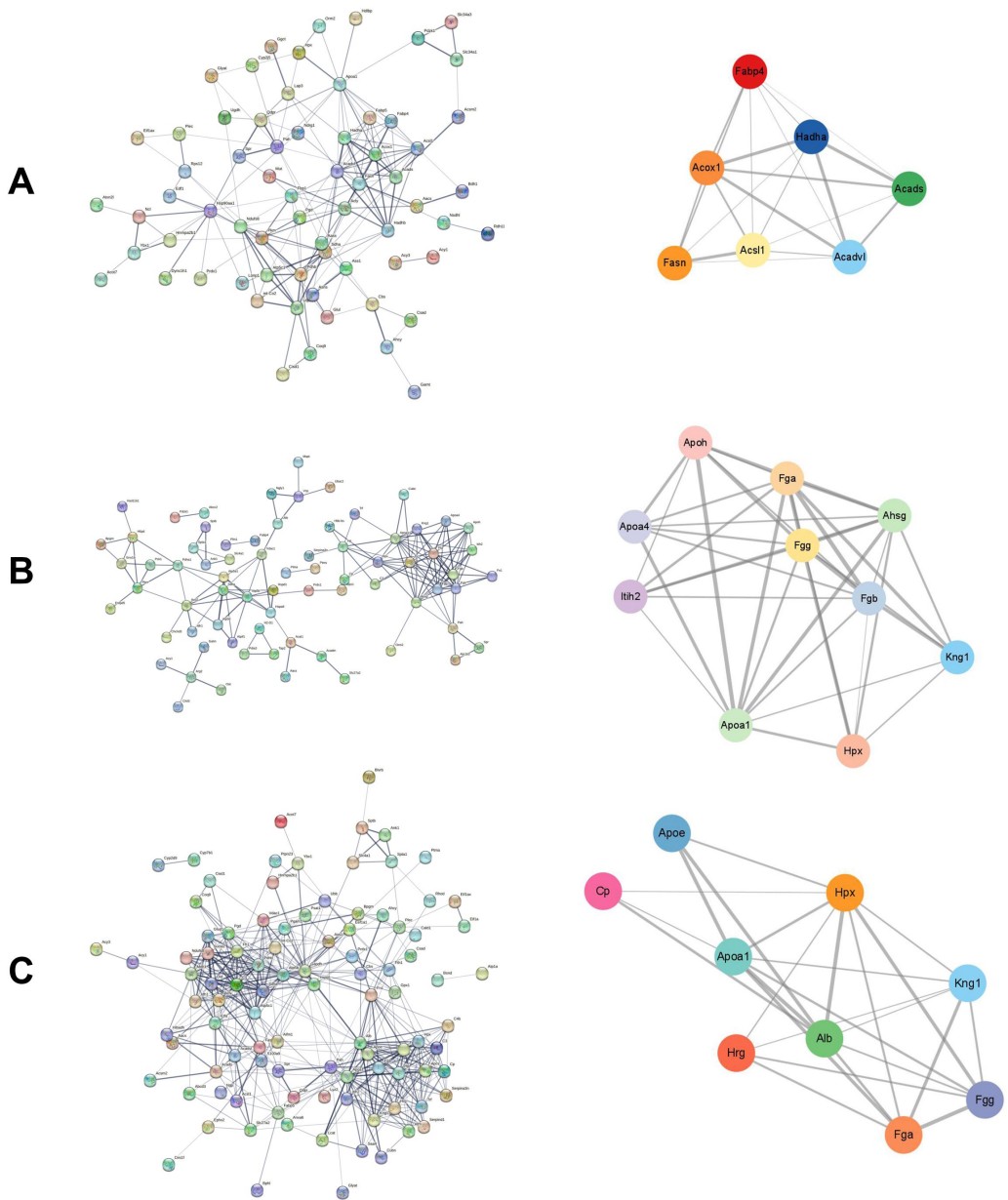

**Figure 3** **PPI network and module analysis.** (A) The PPI network and key modules of DEPs in the cisplatin group *vs* the control group. (B) The PPI network and key modules of DEPs in the cisplatin+SIS3 group *vs* cisplatin group. (C) The PPI network and key modules of DEPs in the SIS3 group *vs* control group. The straight lines (edges) in the module represent the interactions between proteins. The thicker the lines, the stronger the interaction between them.

## PRM validation of key proteins

To verify the accuracy of proteomics data, we selected four overlapped DEPs with key functions, namely NDRG1, FABP4, ACSL1 and Hemopexin, and determined their expression levels using PRM quantitative analysis. In the iTRAQ analysis results, the expression levels of these proteins significantly changed, and they were closely related

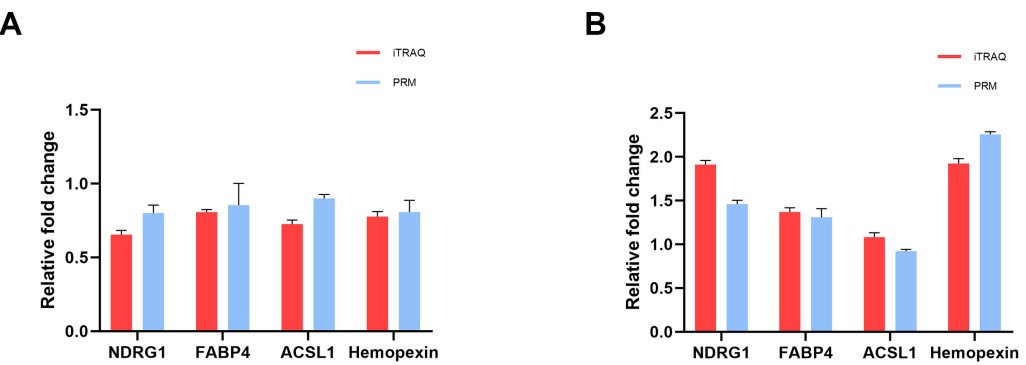

**Figure 4  Quantitative results of iTRAQ and PRM for NDRG1, FABP4, ACSL1, and hemopexin.** (A) cis-platin group *vs* control group. (B) Cisplatin+SIS3 group *vs* cisplatin group. Relative fold change refers to the ratio of two comparative groups.

to characteristics such as lipid metabolism, iron metabolism, and cell proliferation and damage. The RRM validation results showed that these four proteins had a consistent trend with iTRAQ analysis, with a ratio of less than 1 in the cisplatin group *vs* the control group (Fig. 4A), and a ratio greater than 1 in the cisplatin+SIS3 group *vs* the cisplatin group (Fig. 4B), indicating that the iTRAQ results are reliable and can be further analyzed.

## SIS3 reduces lipid accumulation in cisplatin-stimulated mTECs

Renal TECs have high energy consumption, and their energy source is mainly provided by FAO (*Houten et al., 2016*). FAO is inhibited in cisplatin-induced AKI (*Portilla et al., 2002*). The proteomic results of this experiment suggested that Smad3 may regulate lipid metabolism in cisplatin-induced AKI. To further demonstrate this viewpoint, cisplatin was used to stimulate mTECs which were then treated with SIS3. The results showed that cisplatin increased the expression of phosphorylated Smad3 (Figs. 5I and 5J), SCD1 and SREBF1 (Figs. 5D, 5G and 5H), decreased CPT1A and PPARα (Figs. 5D, 5E and 5F), and aggravated a large amount of lipid droplets deposited in mTECs (Figs. 5A–5C), compared with the control group. After treatment with SIS3, the expression of phosphorylated Smad3 (Fig. 5I and 5J), SCD1 and SREBF1 (Figs. 5D, 5G and 5H) decreased, CPT1A and PPARα (Figs. 5D, 5E and 5F) increased, and the accumulation of lipids in mTECs decreased (Figs. 5A–5C). These results indicate that SIS3 reduces lipid accumulation and may promotes FAO in cisplatin-stimulated mTECs. However, it is also possible that the effect is indirect, SIS3 could be protecting cell dysfunction and or cell death and maintaining cellular metabolic health, but not directly affecting FAO.

## SIS3 alleviates oxidative stress in cisplatin-stimulated mTECs

Deficiency of FAO leads to accumulation of free fatty acids in cells, which is an important source of reactive oxygen species (ROS) and one of the important substances causing oxidative stress (*Inoguchi et al., 2000*; *Cury-Boaventura & Curi, 2005*). In hepatocytes and vascular cells, free fatty acids promote ROS production by activating NADPH oxidase through protein kinase C (*Inoguchi et al., 2000*; *Soardo et al., 2011*). We then assessed

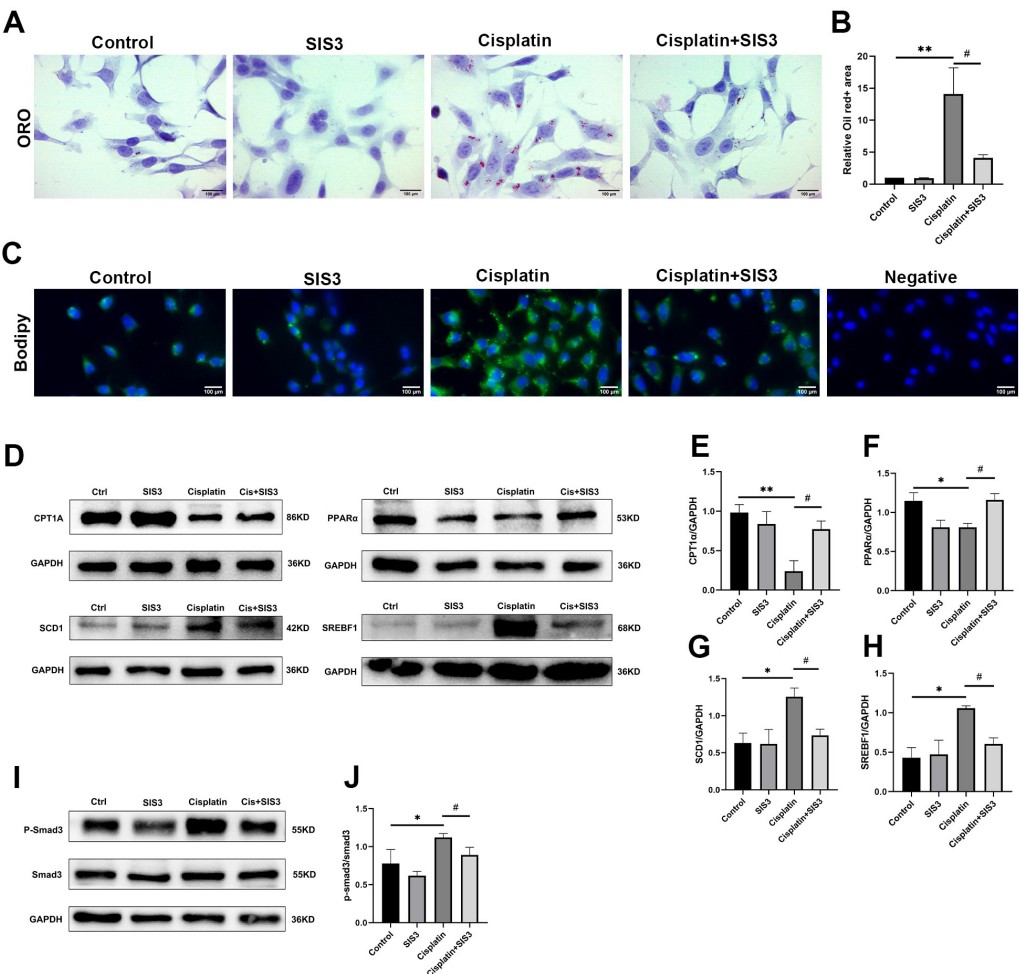

**Figure 5** **SIS3 reduces lipid accumulation and promotes FAO in vitro.** (A–B) Oil Red O staining of mTECs (Bar = 100 μm) and relative quantitative statistical results ($n = 5$). (C) Bodipy493/503 staining of mTECs (Bar = 100 μm). (D–J) Western blot and relative quantitative statistical results ($n = 3$). Each bar represents the mean ± SEM. $*p < 0.05$, $**p < 0.01$, *versus* the control group. $\#p < 0.05$, *versus* the cisplatin group.

cellular oxidative stress by detecting ROS and MDA levels in mTECs. The results showed that cisplatin significantly increased the levels of ROS and MDA, compared with the control group (Figs. 6A, 6B, 6C and 6D). However, SIS3 significantly reduced the levels of ROS and MDA, compared with the cisplatin group (Figs. 6A, 6B, 6C and 6D). These observations indicate that under stimulation by cisplatin, the accumulation of free fatty acids in mTECs leads to an increase in ROS levels, promotes oxidative stress, causes lipid peroxidation, and increases MDA production. Specific inhibition of Smad3 may block this negative effect.

## SIS3 upregulates NDRG1 expression

Based on the iTRAQ and PRM results, we further validated the expression of NDRG1 in mTECs following stimulation by cisplatin and treatment with SIS3 by Western blot and immunohistochemistry. The results showed that NDRG1 was mainly expressed in the

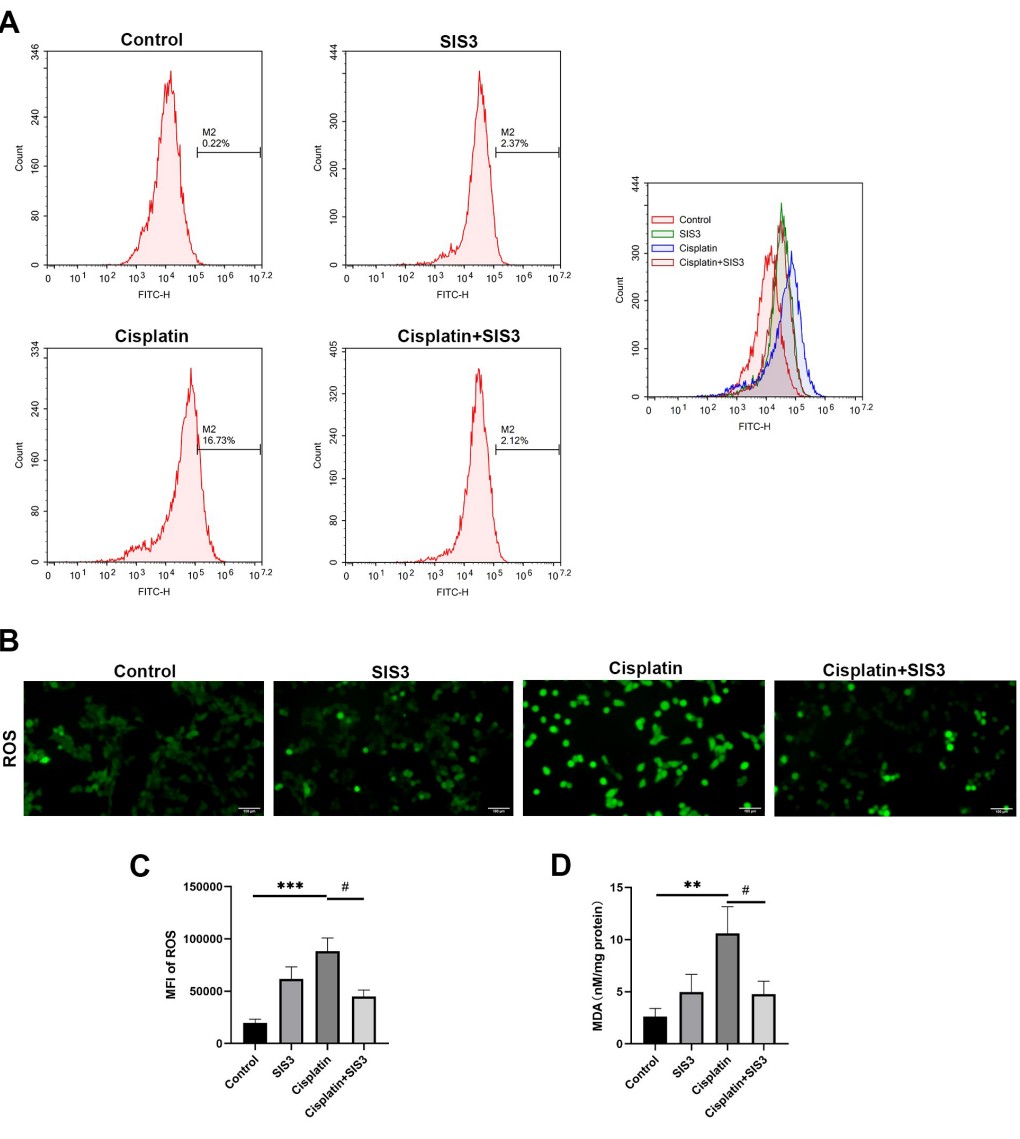

**Figure 6** **SIS3 alleviates oxidative stress *in vitro*.** (A) Following incubation with DCFH-DA, flow cytometry was used to detect the ROS level, using 488 nm excitation wavelength and 525 nm emission wavelength to detect the intensity of DCF fluorescence in real-time. The fluorescence spectrum of DCF is very similar to FITC, and DCF can be detected using FITC parameter settings. The abscissa represents the FITC fluorescence intensity, and the ordinate represents the number of mTECs. (B) After incubation with DCFH-DA, the ROS level was directly observed under an inverted fluorescence microscope (Bar = 100 μm). (C) The average relative fluorescence intensity of ROS in each group of mTECs. (D) The MDA level. Each bar represents the mean ± SEM. **$p < 0.01$, ***$p < 0.001$, *versus* the control group. #$p < 0.05$, *versus* the cisplatin group. $n = 5$.

cytoplasm of renal TECs in mouse kidneys (Fig. 7C). Cisplatin decreased the expression of NDRG1 *in vivo* (Figs. 7C and 7D) and *in vitro* (Figs. 7A and 7B), compared with the control group. Following treatment with SIS3, the expression of NDRG1 increased (Figs. 7A–7B, 7D), compared with the cisplatin group.

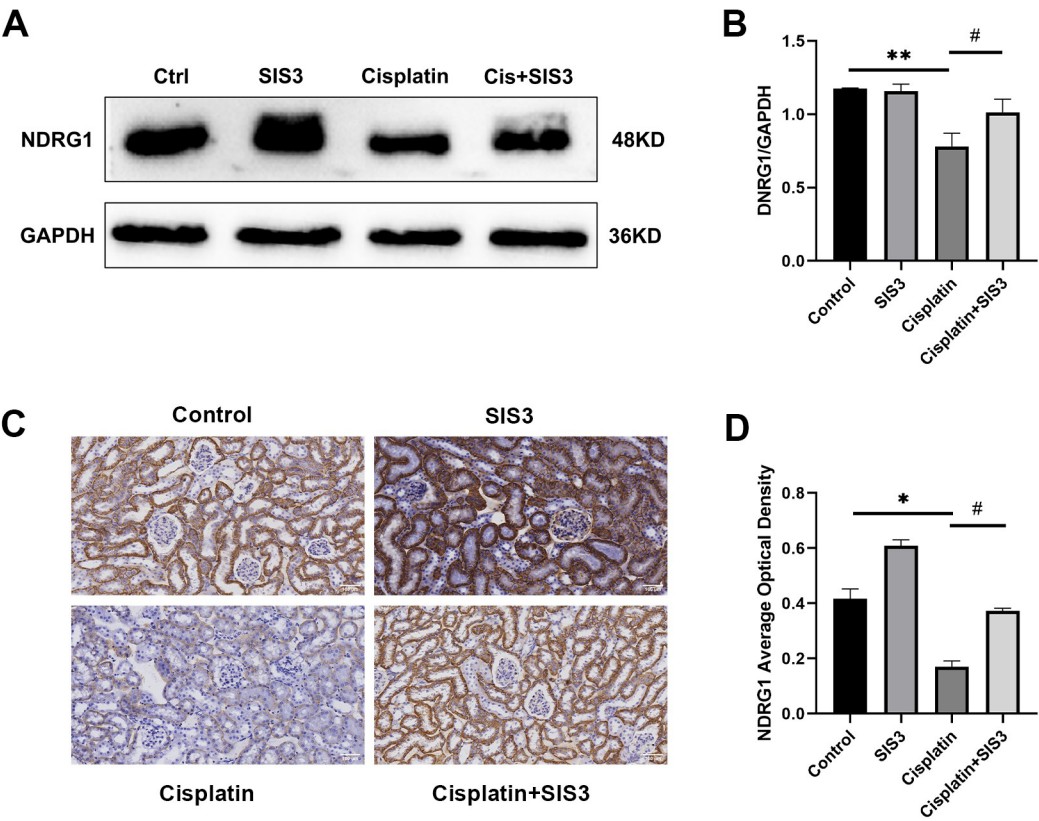

**Figure 7** **SIS3 upregulates NDRG1 expression.** (A–B) Western blot and relative quantitative statistical results of NDRG1 ($n = 3$). (C–D) Immunohistochemical (Bar = 100 μm) and relative quantitative statistical results of NDRG1. Each bar represents the mean ± SEM. $*p < 0.05$, $**p < 0.01$, *versus* the control group. $\#p < 0.05$, *versus* the cisplatin group. $n = 3$.

## DISCUSSION

Proteomic research based on iTRAQ-PRM technology is a newly emerging high-throughput technology for sensitively capturing protein changes, and is also seen as crucial in the discovery of key protein molecules that affect pathological processes (*Latterich, Abramovitz & Leyland-Jones, 2008*). In this study, iTRAQ combined with HPLC-MS/MS and PRM techniques were used to analyze DEPs in mice with cisplatin-induced AKI in the presence or absence of SIS3. Among the 4,787 proteins identified, DEPs were selected based on two criteria (up 1.2-fold or down 0.83-fold, $p < 0.05$). The DEPs were then subjected to GO functional annotation analysis, KEGG pathway enrichment analysis, and PPI network analysis.

Our prior research established that FAO plays a crucial role in modulating cisplatin-induced AKI *via* Sirt3 (*Li et al., 2020*). Additionally, SIS3 is shown to regulate the cell cycle, offering protection against apoptosis and promoting cellular regeneration, thereby mitigating the effects of cisplatin-induced AKI (*Huang et al., 2022*). In addition, it was also recently found that Smad3 can inhibit FAO through the key transcription factor PPARGC1A that regulates FAO in human renal fibrosis samples (*Kang et al., 2015*).

Smad3 deficiency in mice promotes FAO and protects against insulin resistance and obesity induced by a high-fat diet (*Tan et al., 2011*). Sekiguchi K also found that TGF-β/Smad3 signaling pathways directly suppress PPARα activity and reduce FAO in cardiac myocytes (*Sekiguchi et al., 2007*). PPARα is a key transcription factor in the FAO process (*Barger & Kelly, 2000*), However, it is still unclear whether Smad3 mediates the occurrence of FAO in cisplatin-induced AKI. In our study, GO and KEGG analysis showed that these pathways enriched by DEPs are highly involved in energy metabolism, including fatty acid metabolism, the citrate cycle, oxidative phosphorylation, organic acid metabolism, cholesterol metabolism, PPAR signaling pathways, and so on. PPI analysis showed that lipid metabolism is the most critical among them. This indicates that Smad3 may regulate fatty acid metabolism in cisplatin-induced AKI.

Based on the above analysis, we constructed mTECs stimulated by cisplatin (5 μmol/L) in the presence or absence of SIS3. We found that cisplatin increased intracellular lipid droplet deposition, caused lipid peroxidation, increased ROS and MDA production, increased Smad3 phosphorylation, decreased the expression of FAO key proteins CPT1A and PPARα, and increased the expression of lipid synthesis-related proteins SREBF1 and SCD1. CPT1A is a key enzyme in the transport of fatty acids into mitochondria (*McGarry & Brown, 1997*), SREBF1 and SCD1 are important molecules in lipid synthesis (*Kim et al., 2000*; *Horton, Goldstein & Brown, 2002*). However, when SIS3 inhibited the phosphorylation level of Smad3 protein, the above reactions were reversed. This indicated that SIS3 alleviated cisplatin-induced disordered FAO.

In addition, we also found that SIS3 upregulated the decline in NDRG1 expression following cisplatin stimulation. NDRG1 is a protein induced under a variety of stress and cell growth regulatory conditions, involving a variety of cellular processes, including cell proliferation, differentiation, and survival (*Melotte et al., 2010*). It is upregulated by cell differentiation signals in various cancer cell lines and plays a multifaceted role in inhibiting carcinogenic signal transduction (*Ellen et al., 2008*; *Melotte et al., 2010*). Previous studies have shown that NDRG1 is involved in regulating lipid metabolism. Sevinsky CJ found that NDRG1 is a key regulator of lipid metabolism in breast cancer cells. NDRG1 inhibits the formation of lipid droplets in breast cancer cells, and silencing NDRG1 will lead to increased incorporation of neutral lipids in cells and fatty acids in lipid droplets (*Sevinsky et al., 2018*). NDRG1 can also limit infection by inhibiting the formation of lipid droplets to block the assembly of hepatitis C virus (HCV) (*Schweitzer et al., 2018*). In contrast, *Zhao et al. (2022)* and *Wang et al. (2019)* found that NDRG1 promotes the formation of intracellular lipid droplets in studies related to rabies virus RABV and porcine reproductive and respiratory syndrome virus PRRSV infection, respectively. Using iTRAQ quantitative proteomics analysis, we found that NDRG1 expression in the cisplatin group decreased, and in contrast, expression increased in the cisplatin+SIS3 group. PRM and Western blot results showed high consistency with iTRAQ results, indicating the accuracy and repeatability of this method. We speculate that NDRG1 may promote FAO in cisplatin-induced AKI, and the specific role of NDRG1 in regulating lipid metabolism may depend on specific cell types or organizational environments. Epithelial mesenchymal transition (EMT) plays a key role in tumor metastasis, and NDRG1 has been shown to interact with Smad3 and

regulate TGF-β induced EMT. *Chen et al. (2012)* found that NDRG1 can reduce p-Smad3, thereby inhibiting the TGF-β/Smad signaling pathway and EMT in HT29 and DU145 cells. Our results are consistent with these studies, as we found that SIS3 can upregulate the decreased expression of NDRG1 following cisplatin stimulation, indicating that NDRG1 may interact with Smad3 to jointly regulate FAO in cisplatin-induced AKI. However, our study is not without its limitations. The precise function of NDRG1 in relation to fatty acid metabolism in the context of acute kidney injury (AKI) remains incompletely resolved. Aspects such as the interplay between NDRG1 and alternative signaling pathways, as well as the identification of upstream and downstream effectors of NDRG1, warrant further elucidation. Investigating these will enhance our understanding of NDRG1's role in AKI, thereby underpinning the theoretical framework for devising more efficacious therapeutic interventions.

## CONCLUSIONS

In conclusion, based on iTRAQ proteomics, our study undertook a thorough analysis of the variations in protein expression within the kidneys of mice subjected to cisplatin-induced AKI, both with and without the administration of SIS3. Our findings indicate that SIS3 contributes to a reduction in lipid accumulation and attenuates oxidative stress within cisplatin-stimulated mTECs. Furthermore, NDRG1 emerges as a potential protein implicated in the regulation of fatty acid metabolism through Smad3 mediation (summarized in Fig. 8). Our research introduces a novel candidate protein and provides fresh perspectives for unraveling the molecular intricacies of fatty acid metabolism dysregulation in the context of cisplatin-induced AKI.

## ACKNOWLEDGEMENTS

We appreciate all participants in this work.

### Funding

The study was supported by the National Natural Science Foundation of China (Grant No. 81700598), the Health Commission of Jiangxi province in China (Grant No. 202130172) and the Traditional Chinese Medicine of Jiangxi Province in China (Grant No. 2020A0002). The funders had no role in study design, data collection and analysis, decision to publish, or preparation of the manuscript.

### Grant Disclosures

The following grant information was disclosed by the authors:
National Natural Science Foundation of China: 81700598.
Health Commission of Jiangxi province in China:  202130172.
Traditional Chinese Medicine of Jiangxi Province in China: 2020A0002.

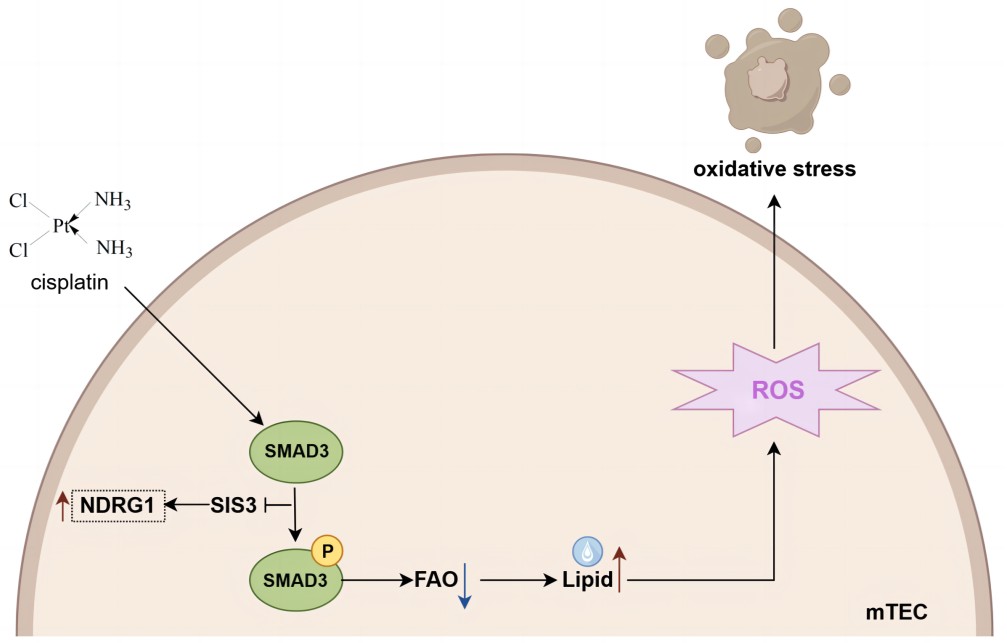

**Figure 8  Mechanistic model of SIS3 improving FAO defects and oxidative stress and regulating NDRG1 in cisplatin-induced AKI.** Figure created in Figdraw (https://www.figdraw.com).

## Competing Interests

The authors declare there are no competing interests.

## Author Contributions

- Jiayan Huang conceived and designed the experiments, performed the experiments, analyzed the data, prepared figures and/or tables, authored or reviewed drafts of the article, and approved the final draft.
- Jian Ye conceived and designed the experiments, performed the experiments, analyzed the data, prepared figures and/or tables, authored or reviewed drafts of the article, and approved the final draft.
- Yi Gao performed the experiments, prepared figures and/or tables, and approved the final draft.
- Yu Wang performed the experiments, prepared figures and/or tables, and approved the final draft.
- Qing Zhao performed the experiments, prepared figures and/or tables, and approved the final draft.
- Tanqi Lou analyzed the data, authored or reviewed drafts of the article, and approved the final draft.
- Weiyan Lai conceived and designed the experiments, performed the experiments, analyzed the data, prepared figures and/or tables, authored or reviewed drafts of the article, and approved the final draft.

## Animal Ethics

The following information was supplied relating to ethical approvals (i.e., approving body and any reference numbers):

The Institutional Animal Care and Use Committee of the First Affiliated Hospital of Nanchang University provided full approval for this research (approval number: CDYFY-IACUC-202308QR037, Nanchang, China).

## Data Availability

The raw data is available in the Supplementary Files.

## Supplemental Information

Supplemental information for this article can be found online at http://dx.doi.org/10.7717/peerj.17485#supplemental-information.

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
