# Peer review of "Identification of proteins related to SIS3 by iTRAQ and PRM-based comparative proteomic analysis in cisplatin-induced acute kidney injury"

_PeerJ, doi:10.7717/peerj.17485_

## Round 0.1 · original submission · Major Revisions

Thank you for submitting to PeerJ. The reviewers were very positive but did note some issues which need to be addressed in full before publication. In particular, please be sure to address copyediting issues, in addition to the scientific issues raised by both reviewers.

**Language Note:** The Academic Editor has identified that the English language must be improved. PeerJ can provide language editing services - please contact us at copyediting@peerj.com for pricing (be sure to provide your manuscript number and title). Alternatively, you should make your own arrangements to improve the language quality and provide details in your response letter. – PeerJ Staff

Reviewer 1 ·

Basic reporting

The manuscript under review presents a robust and well-crafted document with an innovative perspective on using proteomics to identify the mechanism promoters of cisplatin-induced acute kidney injury. The writing is succinct and clear, with minor grammar revision, featuring a well-supported literature introduction and solid data.
1. Grammar- well written text, minor revisions were suggested in the annotated manuscript.
2. Line 316-328 is a duplication of lines 303-315 please delete.
3. Introduction will benefit from giving the name of that antibacterial peptide (line 67), and expand explain the mechanism of the protective effect.
4. Line 69, explain how b2-microglobulin and albumin can be used in early diagnosis, tie it to mechanism.
5. Line 71, explain the reasoning why CHI3L3 and chitinase are most distinctive biomarkers, are they involved in mechanisms or pathogenesis of AKI?

Experimental design

Experimental design- nicely outlined primary research within Aims and Scope of the journal, the Research question is well defined (line 105-112), data presented answered the authors specific hypothesis and had significant predictive benefit which is both relevant & meaningful. Methods were described well and were relevant to the data presented, please see comments in the manuscript text.
6. Figures 1,2,3- offer high resolution images, as of now the writing and figure annotation small and hard to read.
7. Results line 297-299, long literature section should be incorporate in the introduction, if needed state in results in short.
8. Discussion will benefit from additional note on future experiments, or how these findings can help promote new therapeutics for AKI.

Validity of the findings

Validity of the Findings- Conclusions are well stated, linked to original research question and well supported with literature.

Annotated reviews are not available for download in order to protect the identity of reviewers who chose to remain anonymous.

Reviewer 2 ·

Basic reporting

1. This is a well written paper but with occasional typos. For example, line 285, ‘SIS3 reduces lipid accumulationin cisplatin-stimulated mTECs’.

2. The structure of the paper is good with sufficient raw data. But it would be easier for comprehension if the authors could add a summary diagram of the study.

3. Figure 1-3 are very hard to read in detail. Please improve their resolution.

Experimental design

1. Clinical and pathological indexes indicating the successful mice modeling need to be added. Please add reference to the administration of cisplatin and SIS3 in the method. And the impact of SIS3 on clinical and pathological indexes of control and cisplatin-AKI mice also needs to be added.

2. Why did the authors validate only NDRG1 instead of other DEPs in iTRAQ and PRM? Line 276-278, NDRG1 appeared for the first time in the study, by “selected 4 overlapped DEPs with key function”. why did the authors select the 4 overlapped DEPs (NDRG1, FABP4, ACSL1, Hemopexin) while 11 overlapped DEPs have been shown among groups in Figure 1C? Please add the possible reason.

3. Figure 1, it would provide more information of the proteomics if the top 5 or top 10 DEPs could be highlighted in the volcano and heatmap diagram, and the Venn diagram could be divided by up-regulated DEPs and down-regulated DEPs.

Validity of the findings

1. The paragraph of “PPI Network analysis” and Figure 3 demonstrated that the core DEPs in SIS3 or Cisplation+SIS3 were related to cholesterol metabolism. However, the downstream mechanical study focused on lipid accumulation and fatty acid oxidation in general. Have the authors detected the expression of cholesterol metabolism related proteins among groups (for example, SREBP2, LXR)?

3. What is the relationship between the paragraph of “SIS3 upregulates NDRG1 expression” and the previous two paragraphs illustrating SIS3 reduced lipid accumulation and subsequent oxidative stress? How did the authors get the conclusion that "SIS3 alleviates oxidative stress, reduces lipid accumulation and promotes fatty acid oxidation through NDRG1 in cisplatin-induced AKI", while only the NDRG1 expression and lipid accumulation phenotypes are examined. Please add related experiments, for example, over-expression of NDRG1 in cisplatin and cisplatin+SIS3 mice.

3. Line 347-348 in Discussion, “Our previous studies confirmed that Sirt3 is involved in regulating FAO to improve cisplatin-induced AKI”, what is the relationship between Sirt3 and this study?

Additional comments

None

---

## Round 0.2 · accepted · Accept

Reviewers confirmed the authors have addressed comments. Please ensure during proofing that all typos are indeed corrected appropriately.

Reviewer 2 ·

Basic reporting

No comment.

Experimental design

No comment.

Validity of the findings

All questions raised were answered to the point. The reviewer appreciate the authors's hard work.

Additional comments

No comment.